# High Sensitivity Electrochemical As (III) Sensor Based on Fe_3_O_4_/MoS_2_ Nanocomposites

**DOI:** 10.3390/nano13162288

**Published:** 2023-08-09

**Authors:** Haibing Hu, Yunhu Hu, Baozhu Xie, Jianxiong Zhu

**Affiliations:** 1Academy of Opto-Electric Technology, Special Display and Imaging Technology Innovation Center of Anhui Province, National Engineering Laboratory of Special Display Technology, State Key Laboratory of Advanced Display Technology, Collaborative Innovation Center of Advanced Display Technology, Anhui Key Laboratory of Advanced Imaging and Display Technology, Opto-Electric Display Industry Innovation Center, Anhui Province Key Laboratory of Measuring Theory and Precision Instrument, School of Instrument Science and Optoelectronics Engineering, Hefei University of Technology, Hefei 230009, China; m15715608206@163.com (Y.H.); xie_bz@163.com (B.X.); 2School of Mechanical Engineering, Southeast University, Nanjing 211189, China; 3Engineering Research Center of New Light Sources Technology and Equipment, Ministry of Education, Nanjing 211189, China

**Keywords:** As (III) detection, electrochemical analysis method, electrochemical sensors

## Abstract

Currently, heavy metal ion pollution in water is becoming more and more common, especially As (III), which is a serious threat to human health. In this experiment, a glassy carbon electrode modified with Fe_3_O_4_/MoS_2_ nanocomposites was used to select the square wave voltammetry (SWV) electrochemical detection method for the detection of trace As (III) in water. Scanning electron microscopy (SEM) and X-ray diffraction (XRD) showed that Fe_3_O_4_ nanoparticles were uniformly attached to the surface of MoS_2_ and were not easily agglomerated. Cyclic voltammetry (CV) and electrochemical impedance spectroscopy (EIS) showed that Fe_3_O_4_/MoS_2_ has higher sensitivity and conductivity. After optimizing the experimental conditions, the Fe_3_O_4_/MoS_2_-modified glassy carbon electrode exhibited high sensitivity (3.67 μA/ppb) and a low detection limit (0.70 ppb), as well as excellent interference resistance and stability for As (III).

## 1. Introduction

Pollution of the environment has always been a pressing issue in the contemporary domestic and international context. Multiple environmental problems, including soil, air, radioactive elements, and water pollution, exist simultaneously, and these pollutants are always a threat to the human living environment, with heavy metals and heavy metal-like substances in the water environment posing the most direct and obvious threat to humans. Numerous heavy metals, including nickel (Ni), chromium (Cr), lead (Pb), and mercury (Hg), are regarded as human carcinogens [1]. Heavy metals in water resources contribute to ecological cycles, penetrate and enrich the food chain, and cause harm to the food chain’s participants. As is a heavy metal-like element that is extensively distributed in soils, minerals, and aquatic and atmospheric environments. Arsenic rates are 20th relative to other elements in the earth’s crust, 14th relative to other elements in the marine environment, and 12th relative to other elements in human systems [2]. Similar to other metal contamination, trivalent arsenic (As (III)), found in natural waters, poses the greatest threat to human survival, and arsenic and its compounds are mobile in the environment. In the 1970s, the US National Pesticide Monitoring Program tested for residues of mercury, arsenic, lead, cadmium, and selenium in fish and found that in over 95% of the combined samples, residues of all metals were detected, and this number is increasing every year [3]. Therefore, it is necessary to investigate an efficient method for detecting arsenic (As (III)) in the aquatic environment.

Traditional techniques for detecting As (III) include atomic emission spectrometry (AES), atomic absorption spectrometry (AAS), fluorescence analysis, chromatography, and biosensor methods; however, electron tongues have recently been developed for heavy metal detection [4]. These methods, despite being extremely precise, require specialized laboratories, costly apparatus, and specialized techniques to implement. This has hampered their development of large-scale field tests and swift detection. In contrast, electrochemical methods have distinct advantages for detecting heavy metal ions, including low cost, excellent selectivity, high sensitivity, and ease of use [5]. After an extended period of development, electrochemical detection techniques have reached a relatively high level of maturity and are used in numerous applications for the detection of heavy metals. Electrochemical techniques are divided into amperometric, voltammetric, potentiometric, impedance measurement, coulometric, and electrochemiluminescent methods based on the type of electrical signal [6]. Voltammetry, the most advanced electrochemical analysis technique, is further subdivided into square wave voltammetry, differential pulse voltammetry, and cyclic voltammetry based on the type of voltage applied during detection [7]. In electrochemical decontamination, square wave voltammetry can suppress background currents more effectively, resulting in more accurate test results [8]. In addition to selecting an appropriate electrochemical detection method, a negative voltage enrichment method is typically employed prior to the start of the assay to deposit the target ions on the electrode surface, thereby increasing the sensitivity of the process. Currently, the most prevalent electrode materials for arsenic analysis are precious metals such as gold [9,10]. Gold and arsenic can form bimetallic As–Au alloys. Although these noble metals are used to modify electrodes, they are unsuitable for mass assays due to their high cost and the need for manipulation under strong acid conditions, which can generate toxic arsenic vapors. To compensate for this deficiency, it is necessary to select new cost-effective materials [11].

In recent years, metal oxide nanomaterials with outstanding catalytic activity and adsorption capacity have been creatively introduced into the voltammetric response of inorganic arsenic for the detection of arsenic [12,13,14]. Fe_3_O_4_ has garnered considerable interest as a low-cost, eco-friendly, and simple-to-prepare material with optimal catalytic and adsorption activity, as well as high sensitivity and rapid response to arsenic by electrochemical methods [5]. Nanoparticles are extensively utilized because of their high surface area and charge density [15]. By modifying electrodes with graphene and its derivatives, previous research has attained very high sensitivity and detection limits. When graphene and its derivatives are paired with other materials, various contaminants such as copper, lead, chromium, and nickel [16], aquatic organism-toxic chlorophenols [17], melatonin and pyridoxine in plants [18], and ascorbic acid, dopamine, and uric acid can be detected [19]. However, the two-dimensional (2D) material molybdenum disulfide (MoS_2_), with its layered structure similar to graphene, serves an important role as a semiconductor, solid lubricant, catalyst, and more [20]. In the past few years, scientists have discovered that MoS_2_ has a lower charge transfer resistance than metal oxides and high catalytic activity. In general, its surface has numerous protruding sites resembling flower bulbs; this distinctive appearance will help to increase the specific surface area and strengthen its ability to adsorb arsenic ions [21]. In addition, its inexpensive and readily available raw materials and high yield make it a promising graphene substitute. 

In this paper, in order to detect trace arsenic ions in the aqueous environment by electrochemical means, we chose Fe_3_O_4_/MoS_2_ composites to measure the concentration of As (III) in water, considering the reliability, reproducibility, and low cost of detection. A novel and simple composite method was utilized to prepare Fe_3_O_4_/MoS_2_ nanocomposites. By optimizing the experimental parameters, the best experimental results were obtained, and interference immunity and stability experiments were carried out to obtain an electrochemical sensor with high sensitivity and a low limit of detection.

## 2. Materials and Methods

### 2.1. Materials/Chemicals

Triethanolamine (C_6_H_15_NO_3_), arsenic standard solution (As (III)), an acetic acid-sodium acetate buffer solution, and a phosphate buffer solution were purchased from Hefei Beldi Chemical Technology Co. (Anhui, China). Ferric chloride hexahydrate (FeCl_3_·6H_2_O), sodium molybdate dihydrate (Na_2_MoO_4_·2H_2_O), L-cysteine (C_3_H_7_NO_2_S), potassium ferricyanide (K_3_[Fe(CN)_6_]), potassium chloride (KCl), and the interfering ion-detecting reagents were purchased from Aladdin Biochemistry and Technology Co. The deionized water used in the experiments was prepared by an ultrapure water machine. All the experiments were carried out at room temperature.

### 2.2. Instrumentation

All electrochemical tests were carried out on a CHI760e (ChenHua Instruments Co., Shanghai, China) electrochemical workstation equipped with a three-electrode system [22]. The bare or modified glassy carbon electrode was the working electrode, the platinum wire electrode was the counter electrode, and the calomel electrode containing saturated KCI was the reference electrode. The morphology and structure of the materials were characterized with a field emission scanning electron microscope (Gemini 500 SEM microscope, Carl Zeiss, Oberkochen, Baden Wurttemberg, Germany). The material was characterized by XRD using an X-ray diffractometer (Bruker D8, Ettlingen, Germany). The molecular structure and atomic valence of the material were analyzed using X-ray photoelectron spectroscopy (XPS, ESCALAB250Xi, Thermo, Waltham, MA, USA).

### 2.3. Synthesis of Fe_3_O_4_/MoS_2_ Nanocomposites

The preparation method of Fe_3_O_4_ nanoparticles in Fe_3_O_4_/MoS_2_ nanocomposites was adapted from a previous preparation scheme. Furthermore, because the preparation process would involve high temperature and high-pressure reactions in an autoclave, which could be explosive, the composites were prepared by a method in which we provided the preparation scheme, and a third party prepared the composites on our behalf.

First, 0.54 g of ferric chloride hexahydrate (FeCl_3_·6H_2_O) and 4 mL of triethanolamine (TEA) were dissolved in 40 mL of deionized water to produce an alkaline solution of Fe (III)-TEA. Complete dissolution required vigorous stirring on a magnetic stirrer at room temperature. Then, this solution was transferred to a Teflon-lined autoclave and heated at 180 °C for 1.5 h. Finally, the black magnetic product in the cooled autoclave was collected using a magnet, and the collected product was also cleaned thoroughly three times with ethanol and distilled water. After the final magnetic collection, the final product was dried in a vacuum oven at 80 °C for 0.5 h, which was the prepared ferric oxide (Fe_3_O_4_) nanoparticles.

Molybdenum disulfide (MoS_2_) was synthesized in the following manner: 0.3 g of sodium molybdate dihydrate (Na_2_MoO_4_·2H_2_O) was added to a beaker containing 40 mL of deionized water, the solution was sonicated for 20 min, and the pH was adjusted using 0.1 mol per liter of sodium hydroxide (NaOH) solution, which was added dropwise to a pH of 6.5. The solution was then diluted to a volume of 80 mL with deionized water, in which 0.8 g of L-cysteine was dissolved. This was then transferred to a Teflon-lined 100 mL autoclave and heated for 24 h at 240 °C. Similar to the preparation of Fe_3_O_4_ nanoparticles, the black precipitate in the reactor was collected by centrifugation after cooling to room temperature, and the precipitate was washed and desiccated prior to collection; the collected product was molybdenum disulfide.

Lastly, the two substances were physically combined by dispersing 0.2 g of molybdenum disulfide and 0.5 g of ferric tetroxide in 60 mL of deionized water, mechanically stirring the solution for 1 h to ensure thorough mixing, transferring the solution to an autoclave, and heating it at 180 °C for 18 h. Finally, the Fe_3_O_4_/MoS_2_ nanocomposite was obtained by centrifuging and drying the final product.

### 2.4. Preparation of the Fe_3_O_4_/MoS_2_ Nanocomposite Modified Electrode

The glassy carbon electrodes were polished with 1.0 μm, 0.3 μm, and 0.05 μm aluminum oxide powders, and the polished electrodes were subjected to cyclic voltammetry tests using an electrochemical workstation in a mixture of 5 mmol/L K_3_[Fe(CN)_6_] and 0.1 mol/L KCl at a voltage range of −0.2 V to 0.8 V. The polished electrode was considered clean if the peak potential difference of the working electrode was within 100 mV; otherwise, the polishing process was repeated. Then, the working electrode was cleaned with deionized water and dried in a clean room to allow natural drying until the surface of the glassy carbon electrode was free of deionized water. Take 20 mg of the prepared Fe_3_O_4_/MoS_2_ nanocomposite, pour it into 10 mL of deionized water, and configure it into a suspension of 2 mg per ml. Make sure that the suspension of the Fe_3_O_4_/MoS_2_ nanocomposite is concentrated at the center of the glassy carbon electrode; otherwise, it will have a significant effect on the test results if it is scattered at the edge of the glassy carbon electrode. The suspension needs to be sonicated for 10 min before taking the suspension dropwise to make the Fe_3_O_4_/MoS_2_ nanocomposite uniformly distributed in the deionized water [23]. The 7 μL of the suspension was taken with a pipette gun and dropped onto the center of the pretreated glassy carbon electrode, then placed vertically in a dust-free chamber and dried at room temperature for 9–10 h. When the deionized water evaporated completely, the Fe_3_O_4_/MoS_2_ nanocomposites were attached to the surface of the glassy carbon electrode, and the working electrode was prepared.

### 2.5. Electrochemical Measurements

The electrochemical measurements in this experiment were carried out by a three-electrode configuration system of an electrochemical workstation. In the electrochemical sensing experiments, bare and modified electrodes were measured by cyclic voltammetry (CV) and electrochemical impedance spectroscopy (EIS) under a mixture of 5 mmol/L K_3_[Fe(CN)_6_] and 0.1 mol/L KCl, respectively. As (III) was measured in a buffer solution. It was first deposited at −0.8 V for 160 s, followed by subsequent measurements by square wave voltammetry, and finally desorbed at 0.8 V for 160 s. The square wave voltammetry parameters were set to detect a potential range of −0.6–0.6 V, a potential increase of 0.004 V, an amplitude of 0.025 V, and a frequency of 40 Hz for the square wave.

## 3. Results 

### 3.1. Characterization of Fe_3_O_4_/MoS_2_ Nanocomposites

Figure 1a depicts the scanning electron microscope image of Fe_3_O_4_ nanoparticles. It can be seen from the graph that the average particle size of Fe_3_O_4_ nanoparticles is less than 20 nm, and the majority of them are concentrated around 10 nm, with an aggregation tendency. This pertains to the magnetic properties of Fe_3_O_4_ nanoparticles and indicates that Fe_3_O_4_ particles smaller than 20 nm have been successfully prepared. The SEM characterization of Fe_3_O_4_/MoS_2_ is shown in Figure 1b. From the figure, it can be seen that MoS_2_ is a layered 2D material with many flower spherical protruding sites on its surface, which are ideal attachment sites for Fe_3_O_4_ nanoparticles, making the agglomeration of Fe_3_O_4_ nanoparticles loaded on the surface of MoS_2_ reduced. It can be seen that a large number of Fe_3_O_4_ nanoparticles are attached to the protruding sites on the MoS_2_ surface, and the agglomeration phenomenon is much weaker compared to Figure 1a. This will have a positive effect on the adsorption of arsenic ions in the subsequent assay.

The SEM-EDS elemental mapping image of Fe_3_O_4_/MoS_2_ composites is shown in Figure 1c, obtained by face scanning with X-ray energy spectrometry of the hot field projection scanning electron microscope. When the electron beam scans the surface of Fe_3_O_4_/MoS_2_ composite samples, the distribution of elements on the surface of the specimen will be shown in different brightness or color distributions, which can qualitatively analyze the surface of the sample elements. It can be seen from Figure 1c that four elements, Fe, O, Mo, and S, are present and only present in the sample, but the mapped images of Fe and O elements are less bright compared to Mo and S elements because the Fe_3_O_4_ nanoparticles are uniformly attached to the prominent sites on the surface of the MoS_2_ material and their surface content is relatively small. However, all four elements, Fe, O, Mo, and S, are uniformly distributed on the selected observation surface, which also demonstrates the more successful compounding of Fe_3_O_4_ nanoparticles with the two-dimensional layered MoS_2_ material. Figure 1d is the EDS diagram of the Fe_3_O_4_/MoS_2_ nanocomposite. We can clearly see the relevant elements, mainly Fe, O, Mo, and S. There are no other interfering elements mixed in during the manufacturing process, which also proves that the preparation of Fe_3_O_4_/MoS_2_ nanocomposite was successful.

The XRD profiles of Fe_3_O_4_/MoS_2_ nanocomposites and Fe_3_O_4_ materials after characterization are shown in Figure 2a. There are several distinct diffraction peaks in the XRD characterization curves of Fe_3_O_4_/MoS_2_ nanocomposites, the positions of which can be corresponded to the positions of diffraction peaks in the XRD characterization curves of Fe_3_O_4_ (PDF#77-1545) material, which are located at 2θ = 18.285 (111), 35.426 (311), 43.053 (400) 62.520 (440), and 56.936 (511) for the XRD diffraction peaks (surfaces) of the Fe_3_O_4_ material. Several other apparent diffraction peak positions can correspond to those of the MoS_2_ (PDF#73-1508) material, namely the diffraction peaks (faces) of MoS_2_ at 2θ = 14.390 (002), 32.803 (100), 39.651 (103), and 49.835 (105), which also prove the Fe_3_O_4_/MoS_2_ nanocomposite successful synthesis as well as composite.

The full-spectrum scan of the Fe_3_O_4_/MoS_2_ nanocomposite is shown in Figure 2b, which shows the main Mo 3d, S 2p, Fe 2p, and O 1s characteristic peaks, which indicates that the sample prepared in this paper contains four elements, namely Fe, O, Mo, and S.

Figure 3a–d depicts the high-resolution X-ray photoelectron spectroscopy of Fe_3_O_4_/MoS_2_ nanocomposites S 2p, Mo 3d, O 1s, and Fe 2p, respectively. In this paper, peak splitting is used to process the original scanning data. The outer scatter curve represents the original scanning data of X-ray photoelectron spectroscopy, whereas the outer red curve represents the outer envelope fitted with each minor peak following peak splitting. Indirectly, the degree of fit between the secondary curve and the original data can demonstrate peak splitting’s rationality. Figure 3a depicts the spectrum of fine Fe 2p. The approximate proportional relationship between Fe^2+^ (approximately 711.04 eV) and Fe^3+^ (approximately 713.47 eV) in the Fe_3_O4/MoS_2_ composite reveals that Fe^2+^ has a slightly greater proportion than Fe^3+^. Mo^5+^ (approximately 229.65 eV) is the most prominent peak when compared to Mo^4+^ (approximately 228.8 eV), as seen in the fine spectrum of Figure 3d and Mo 3d. When molybdenum disulfide is combined with ferric oxide, the ferric iron in ferric oxide receives electrons from the active tetravalent molybdenum in molybdenum disulfide, resulting in the formation of Fe^3+^ + Mo^4+^→Fe^2+^ + Mo^5+^. Furthermore, tetravalent molybdenum has greater reducibility than divalent iron. In subsequent investigations, this property will indirectly promote the conversion of As (III) to As (0), which is conducive to the enrichment of arsenic. Figure 3b also displays the fine spectra of S 2p, which correspond to the S 2p_1/2_ orbital (approximately 163.64 eV) and the S 2p_3/2_ orbital (approximately 162.45 eV), respectively.

### 3.2. Electrochemical Characterization of Fe_3_O_4_/MoS_2_ Electrodes

To evaluate the electrochemical performance of the modified glassy carbon electrodes, two electrochemical methods, cyclic voltammetry (CV) and electrochemical impedance spectroscopy (EIS), were chosen for the electrochemical characterization of glassy carbon electrodes modified with different materials and bare electrodes, respectively [24,25].

A mixture of 5 mmol/L K_3_[Fe(CN)_6_] and 0.1 mol/L KCl was selected as the detection solution for the electrochemical characterization, and the [Fe(CN)_6_]^3−^ in the mixture was used as the redox probe to characterize the electrochemical properties of each material modified electrode, as shown in Figure 4a for the cyclic voltammetric characterization of different materials modified electrodes. The corresponding parameters in the cyclic voltammetry process are set as follows: the number of scanning turns is three, the high potential is 0.8 V, the low potential is −0.2 V, the sampling interval is 0.001 V, the scanning rate is 0.05 V/s, and the rest of the parameters can be referred to the default settings. It can be seen that in the cyclic voltammetry test of the bare electrode, there is a pair of redox peaks with essentially the same peak height and symmetrical position; compared to the cyclic voltammetry test of the Fe_3_O_4_ material-modified electrode, the peak current decreases and the potential difference (Δp) at the redox peaks increases. This is because the conductivity of Fe_3_O_4_ nanoparticles is not as excellent as that of the glassy carbon electrode, which will hinder the transfer of electrons on the electric shock and is prone to agglomeration, resulting in a reduced electrochemical active area. The peak current of the cyclic voltammetry curve of the MoS_2_-modified electrode is significantly increased compared to the bare electrode, which is attributed to the prominent active sites on the surface of the MoS_2_ material, but it is also less conductive. The peak currents of cyclic voltammetry curves of Fe_3_O_4_/MoS_2_ nanocomposites were all higher than those of electrodes modified with single materials, and the potential difference (p) at the redox peaks was lower than that of the uncoated electrode. This is because the MoS_2_ material provides sufficient active sites for Fe_3_O_4_ nanoparticle co-adhesion, which reduces the agglomeration of Fe_3_O_4_ nanoparticles, and the two play an excellent synergistic function in enhancing electrical conductivity. This improves the response of electrochemical sensors based on Fe_3_O_4_/MoS_2_ nanocomposites extraordinarily well.

As shown in Figure 4b regarding the electrochemical impedance spectra at bare electrodes, Fe_3_O_4_, MoS_2,_ and Fe_3_O_4_/MoS_2_-modified electrodes, respectively, the frequencies tested were set to 0.01–10^6^ Hz, and as with cyclic voltammetry, a mixture of 5 mmol/L K_3_[Fe(CN)_6_] and 0.1 mol/L KCl was chosen as the detection solution for the electrochemical impedance spectroscopy tests. The straight line part of the electrochemical impedance spectrum corresponds to the low-frequency part of the detection, corresponding to the diffusive transport of ions, while the half-circle part at the front end corresponds to the mid-high frequency part of the detection, indicating the resistance during charge transfer. The impedance spectrum of the Fe_3_O_4_-modified electrode has a larger semicircle, indicating its poor conductivity, while the semicircle of the impedance spectrum of the MoS_2_-modified electrode decreases significantly, and the impedance spectra of Fe_3_O_4_/MoS_2_ nanocomposite modified electrodes are almost a straight line, indicating that the resistance to charge transfer is extremely low and the conductivity is much higher, which again confirms the conclusion of the previous paper that the two Fe_3_O_4_/MoS_2_ play a very effective synergistic role.

To further investigate the relationship between scan rate and the electron dynamics of modified electrodes in cyclic voltammetry, this paper chose to test electrodes modified with different materials at different scan rates, changing the parameter settings as follows: high potential 1 V, low potential −1 V, initial potential 0 V, the number of scan turns was 3, the sampling interval was set to 0.001 V, and multiple scan rates were performed for each different material modified electrode tests, set to 0.01 V/s, 0.02 V/s, 0.03 V/s,…, and 0.1 V/s, respectively, record the peak anode current and the square root of the scan rate for each test, plot a scatter plot with the peak current magnitude as the Y-axis and the square root of the scan rate as the X-axis, and finally perform a linear fit, as shown in Figure 5. 

The equation of the peak current versus the square root of the scan rate for cyclic voltammetry tests with four different electrodes is fitted in the figure as follows:(1)$$Bare:Ip=159.46v^{\frac{1}{2}}+2.03\left(R^{2}=0.999\right)$$
(2)$${Fe}_{3}O_{4}:Ip=116.49v^{\frac{1}{2}}+7.09\left(R^{2}=0.991\right)$$
(3)$$MoS_{2}:Ip=218.64v^{\frac{1}{2}}+4.37\left(R^{2}=0.999\right)$$
(4)$${Fe}_{3}O_{4}/MoS_{2}:Ip=238.49v^{\frac{1}{2}}+0.90\left(R^{2}=0.999\right)$$
where Ip is the peak anode current in the cyclic voltammetry test, and v is the scan rate in the test. From the fitted curves as well as the equations, it can be seen that the anodic peak currents in cyclic voltammetry tests for different material-modified electrodes all increase with increasing scan rates with high confidence. Among the slopes of the fitted curves, the corresponding curve of the Fe_3_O_4_/MoS_2_ nanocomposite-modified electrode has the largest frontal slope, which also proves that the electron transfer rate is the fastest at the surface of this electrode.

### 3.3. Optimization of Analytical Conditions

First, the choice of buffer solution was considered. Through comparing with other papers, it was found that the peak currents were more pronounced in the area adsorption of As (III) in the acetate buffer solution and the phosphate buffer solution. Further experiments were then done on these two buffer solutions, and as shown in Figure 6a, the strongest peak signal for As (III) is found in the acetate buffer solution, while the weaker peak signal for As (III) is found in the phosphate buffer solution. Therefore, the acetate buffer solution was chosen for further experiments.

Next, the pH of the acetic acid-sodium acetate buffer solution was determined, and the relevant experimental results are shown in Figure 6b. It can be seen that the peak response current during square wave voltammetry detection increases and then decreases with the pH of the acetate buffer solution, with the peak current reaching a maximum at pH 6 and a precipitous drop in the peak response current at pH 7. Under acidic conditions, arsenic is more easily ionized and thus more readily enriched on the surface of the glassy carbon electrode. However, the excessive acidity of the buffer solution causes more hydrogen bubbles to be generated on the surface of the glassy carbon electrode during the deposition process, which instead makes it difficult to deposit arsenic ions that would otherwise be deposited on the electrode surface, manifesting as a weakened adsorption capacity for arsenic and affecting the peak current response in subsequent experiments. As with other reports, at lower pH values, the adsorption capacity of As (III) was poor [26,27]. In contrast, at pH values greater than 6, the solubility of arsenate decreased, resulting in a substantial decrease in the peak response current in subsequent tests. Therefore, a buffer solution of sodium acetate at pH 6 was chosen as the buffer solution in the subsequent experiments.

The experimental results are shown in Figure 6c, which shows that the peak current increases and then decreases with the increase of the deposition potential, and the best peak response is reached at the deposition potential of −0.8 V. When the deposition potential is small, the arsenic ions cannot be fully enriched on the glassy carbon electrode in a short time, and the deposition time needs to be lengthened if more desirable results are to be achieved. When the deposition potential exceeds −0.8 V, the presence of ions other than arsenic ions undergoes reduction reactions, which will lead to the subsequent square wave voltammetry detection being affected, and the best response cannot be achieved [28,29]. Therefore, the final deposition potential was determined to be −0.8 V.

As shown in Figure 6d, the peak response current for arsenic ion detection increases gradually with increasing deposition time. However, the increase in peak response current from 120 to 160 s is significantly greater than the increase from 160 to 200 s, resulting in a phenomenon similar to relative saturation due to the deposition and desorption of arsenic ions on the electrode surface. A dynamic equilibrium was reached. Although the longer the deposition time, the larger the peak current signal, the longer the deposition time, the thicker the material on the surface of the glassy carbon electrode, which will not only be detrimental to the desorption of arsenic ions in subsequent experiments but will also affect the sensitivity of the sensor.

### 3.4. Electrochemical Detection of As (III)

The modified glassy carbon electrode is used as an electrochemical sensor to detect arsenic ions in the aqueous environment by square wave voltammetry. Before formal detection, arsenic ions are also deposited by the current–time curve method, where they undergo reduction to arsenic monomers at a deposition potential of −0.8 V and a deposition time of 160 s and are enriched from the buffer solution to the surface of the working electrode, during which the Fe^2+^/Fe^3+^ conversion cycle is involved [30]. A peak response current occurs during the dissolution process, after which a desorption experiment is performed at a desorption potential of 0.8 V for 160 s in order to remove the remaining small amount of arsenic from the surface of the electrode, and the above experimental flow is shown in Figure 7.

After the current–time curve deposition process, arsenic ion solutions with a concentration of 10 ppb were detected by square wave voltammetry under the most suitable conditions using Fe_3_O_4_/MoS_2_ nanocomposite-modified glassy carbon electrode as the working electrode. The parameters of square wave voltammetry were set as follows: a detection potential range of −0.6–0.6 V, a potential increase of 0.004 V, the amplitude of 0.025 V, the frequency of the square wave was 40 Hz, and the detection curve is shown in Figure 8a. At a potential of around 0.07 V, there was a significant peak response current with a value of up to 68.45 µA, which has a more obvious advantage over other literature.

Compared to the bare glassy carbon electrode and the glassy carbon electrode modified with a single material, as shown in Figure 8b, the respective corresponding peak response is significantly weaker than that of the glassy carbon electrode under composite modification. This is because when Fe_3_O_4_ is in contact with MoS_2_ to form a composite material, the electrons from MoS_2_ will be significantly transferred to Fe_3_O_4_. After the drop is attached to the electrode surface, the square wave voltammetry detection process accelerates the transfer of electrons from MoS_2_ to Fe3O_4_, enhancing the previously mentioned Fe^2+^/Fe^3+^ conversion cycle. This cycle will directly and positively affect the aqueous environment in the Fe_3_O_4_/MoS_2_ composite, which will be significantly better than the two alone in terms of detection.

Under optimized experimental conditions, arsenic ion solutions were detected at different concentrations, where the linearity interval was determined to be 1–20 ppb. When the arsenic ion concentration exceeded the larger range of 20 ppb, the peak response current increased, but the non-linearity was enhanced, and its significance as a sensor was lost. The linear relationship between arsenic ion concentration and peak response current is shown in Figure 9, with a fitted correlation coefficient of R_2_ = 0.9988, yielding a sensitivity of 3.67 μA/ppb for the sensor. In the case of K = 3 (where K is the confidence level), the calculated detection limit is 0.7 ppb, much lower than the 10 ppb of arsenic ion concentration in drinking water specified by the World Health Organization. There are also some advantages over other literature, as shown in Table 1.

The strong peak response of Fe_3_O_4_/MoS_2_ nanocomposites in detecting arsenic ions is due to (1) the excellent synergistic effect of MoS_2_ and Fe_3_O_4_, which promotes the Fe^2+^/Fe^3+^ conversion cycle. (2) The MoS_2_ material has a large specific surface area and many prominent sites, providing many attachment sites for the Fe_3_O_4_ material and avoiding many agglomerations of magnetic Fe_3_O_4_ material. (3) The Fe_3_O_4_ material has excellent adsorption performance and selectivity for arsenic ions. (4) Excellent electrochemical properties of the glassy carbon electrode [40].

### 3.5. Interference Resistance, Reproducibility, and Reusability Analysis

As pollution in actual water samples is diverse, different kinds of heavy metal ions may exist in a water sample, which may be deposited together with arsenic ions during the detection process, resulting in deviations in the detection results, which generally make the detection results biased and affect the accuracy of the sensor, so it is necessary to verify the interference resistance of the sensor [41]. In order to simulate the contamination of heavy metals in actual water samples, in this paper, we use the artificial addition of interfering ions and set five interfering ions, Cu^2+^, Pb^2+^, Zn^2+^, Ni^2+^, and Co^2+^, whose concentrations are set to 100 ppb, to coexist with 10 ppb As^3+^ in the water samples to test the degree of influence on the peak response current in the presence of the interfering ions.

Figure 10a shows the detection curve of 10 ppbAs^3+^ when Cu^2+^ with a concentration of 100 ppb is used as an interfering ion. In addition to the peak response corresponding to As^3+^ near 0.07 V, a peak response corresponding to Cu^2+^ appears near −0.06 V, and the peak response of As^3+^ is 69.17 μA. The change is only about 1% compared with the original, which indicates that the presence of Cu^2+^ has little interference with the detection of As (III) by electrochemical sensors based on Fe_3_O_4_/MoS_2_ nanocomposites. In addition, the influence of other common metal ions Pb^2+^, Zn^2+^, Ni^2+^, and Co^2+^ on the peak response current of arsenic detection was also studied, and the results were shown in the test diagram of Figure 10b interfering ions (Cu^2+^, Pb^2+^, Zn^2+^, Ni^2+^, Co^2+^). The difference between the peak response and that of arsenic only was 1.39 μA, 1.24 μA, 0.25 μA, and 1.31 μA, respectively, and the maximum deviation was less than 3%. In addition to the above metal cations, the presence of corresponding concentrations of SO_4_^2−^, Cl^−^, and NO_3_^−^ at the same time will not have a significant impact on the peak response current of arsenic. This shows that the electrochemical sensor based on Fe_3_O_4_/MoS_2_ nanocomposites has excellent selectivity to arsenic and strong anti-interference.

The following experiments were devised to confirm the repeatability of electrochemical sensors based on Fe_3_O_4_/MoS_2_: On the Fe_3_O_4_/MoS_2_-based electrochemical sensor, multiple deposition experiments and dissolution processes were conducted under optimal experimental conditions. Arsenic ions were detected once after each deposition procedure, thereby preventing excessive deposition of arsenic ions on the electrode surface. Figure 11 depicts the maximal current measured numerous times. In the first five experiments, the minimum deviation was only −0.4%, and the maximum deviation was only 2%. However, a 6% deviation was observed in the sixth test relative to the first five, which is a significant deviation and cannot be approved in actual testing. With fewer arsenic ion detections, the electrochemical sensor based on Fe_3_O_4_/MoS_2_ maintains good repeatability, but there is a possibility of significant deviation as the number of detections increases.

Stability is an essential parameter for practical sensors, as it directly impacts the sensor’s service life. The following experiments were devised to test the stability of the sensor described in this paper: The prepared Fe_3_O_4_/MoS_2_ modified glassy carbon electrode was stored in a sterile box at room temperature for 1, 7, and 14 days prior to electrochemical detection of arsenic, the results of which are depicted in Figure 12. After seven days of storage, the maximal current measured was 68.05 μA, and the experimental data showed only a 0.5% deviation from the one-day storage result.After 14 days of long-term storage, the maximal current measured was 66.09 μA, and the deviation was only 3.4%, which is less than 5%.This demonstrates that the electrochemical sensor based on Fe_3_O_4_/MoS_2_ proposed in this paper does not exhibit large short-term test deviation and has a high level of confidence.

After the above series of reliability experimental tests, it is shown that the electrochemical sensor based on Fe_3_O_4_/MoS_2_ has excellent anti-interference and stability when used for arsenic ion detection and has excellent repeatability in a small number of tests. This sensor can be applied to the detection of actual water samples and maintain high reliability in a certain period of time.

## 4. Conclusions

In this study, modified Fe_3_O_4_/MoS_2_ electrodes were chosen for the detection of trace As (III) in water. Fe_3_O_4_/MoS_2_ nanocomposites exhibited excellent electrochemical properties and high sensitivity to trace As (III) with a sensitivity of 3.67 μA/ppb, which is superior to many other materials in the scientific literature [42], and the peak current of the response is far beyond the reach of other papers. In addition, it shows good selectivity for interfering ions, with a maximum error of 3 percent. Moreover, the As (III) test shows satisfactory repeatability and stability, which also determines that the test will not be significantly affected by the number or the duration of tests. The related findings provide a new method for the electrochemical detection of As (III) to prepare materials with good performance.

## Figures and Tables

**Figure 1 nanomaterials-13-02288-f001:**
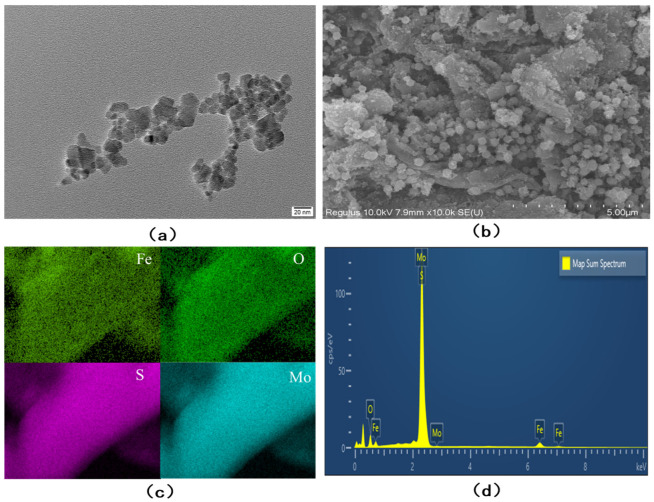
(**a**) TEM scans of Fe_3_O_4_ nanoparticles; (**b**) SEM characterization of Fe_3_O_4_/MoS_2_; (**c**) SEM-EDS elemental mapping images of Fe_3_O_4_/MoS_2_ composites; (**d**) SEM-EDS spectral analysis of Fe_3_O_4_/MoS_2_ composites.

**Figure 2 nanomaterials-13-02288-f002:**
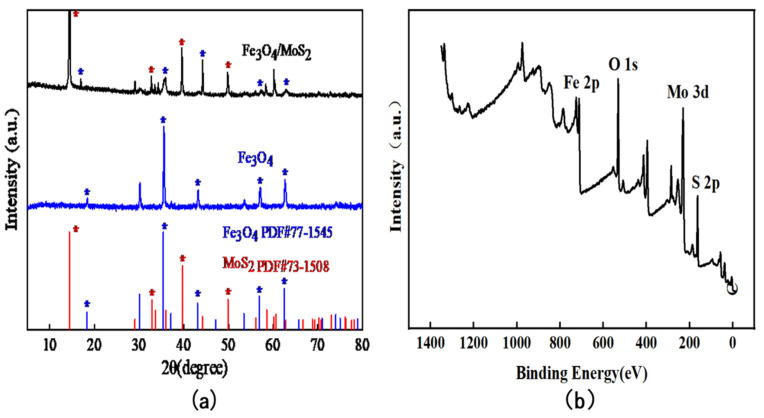
(**a**) XRD characterization of Fe_3_O_4_/MoS_2_ nanocomposites, Fe_3_O_4_ nanoparticles; (**b**) XPS full-spectrum scans of Fe_3_O_4_/MoS_2_ nanocomposites.

**Figure 3 nanomaterials-13-02288-f003:**
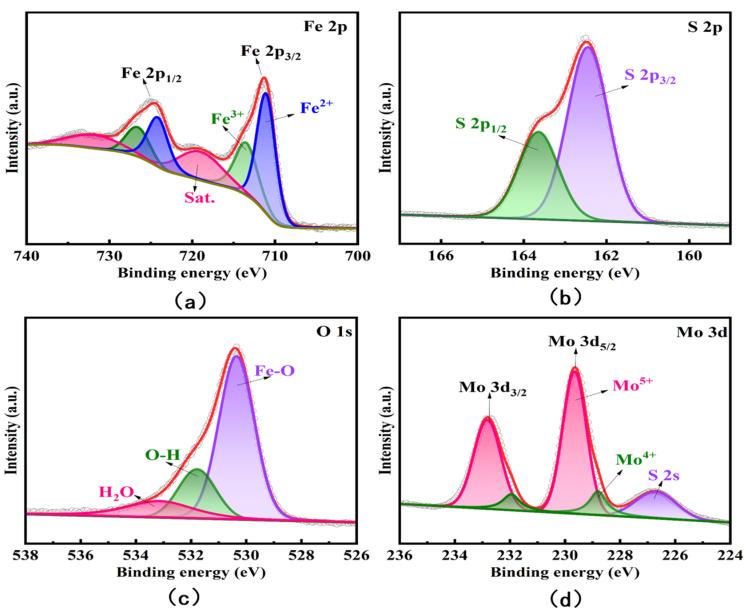
High-resolution XPS spectra of (**a**) Fe 2p, (**b**) S 2p, (**c**) O 1s, and (**d**) Mo 3d of Fe_3_O_4_/MoS_2_ nanocomposites.

**Figure 4 nanomaterials-13-02288-f004:**
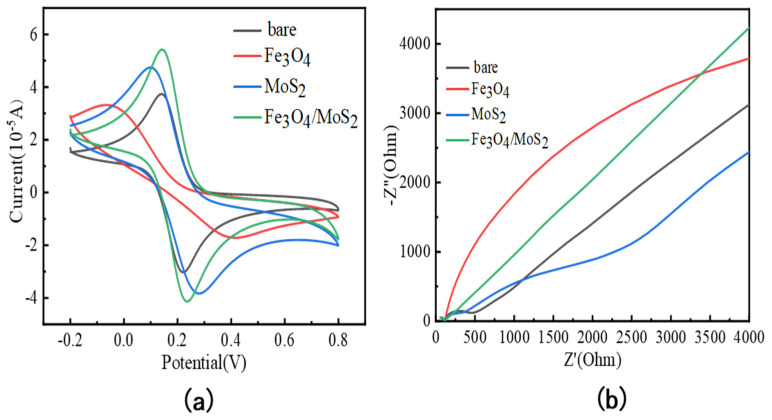
(**a**) Cyclic voltammetric test plots of Fe_3_O_4_/MoS_2_, Fe_3_O_4_, and MoS_2_-modified glassy carbon electrode and bare glassy carbon electrode; (**b**) electrochemical impedance spectra of Fe_3_O_4_/MoS_2_, Fe_3_O_4_, and MoS_2_-modified glassy carbon electrode and bare glassy carbon electrode.

**Figure 5 nanomaterials-13-02288-f005:**
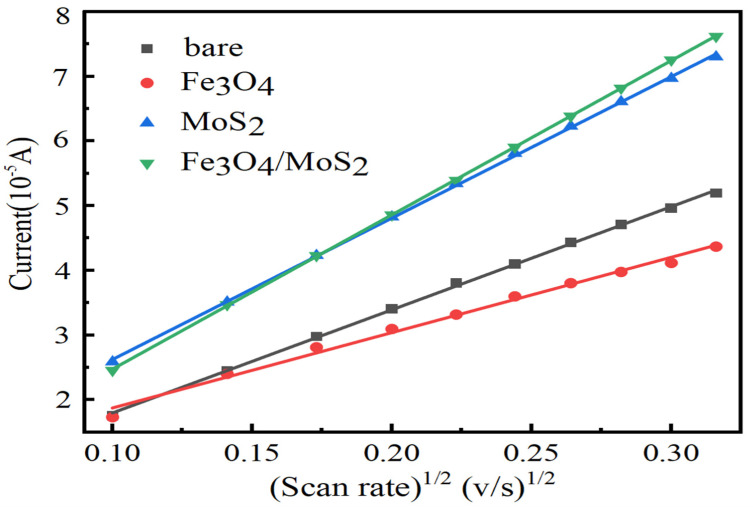
Plot of electron transfer kinetics versus cyclic voltammetric scan rate for Fe_3_O_4_/MoS_2_, Fe_3_O_4_, and MoS_2_-modified glassy carbon electrode and bare glassy carbon electrode.

**Figure 6 nanomaterials-13-02288-f006:**
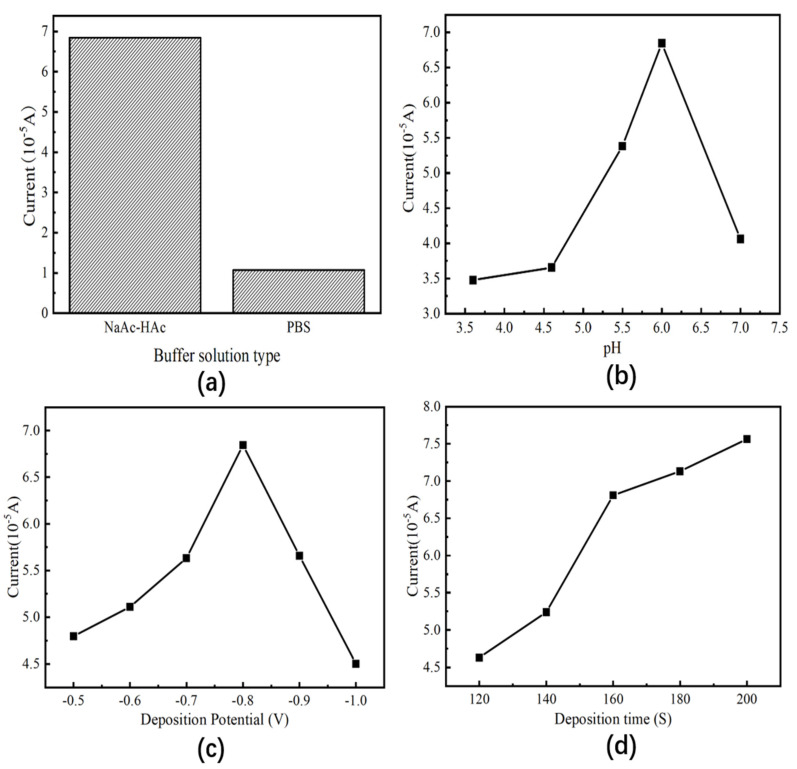
(**a**) Experimental diagram of buffer solution types optimization; (**b**) experimental diagram of pH optimization of buffer solution; (**c**) experimental diagram for optimization of deposition potential; (**d**) experimental diagram for optimization of deposition time.

**Figure 7 nanomaterials-13-02288-f007:**
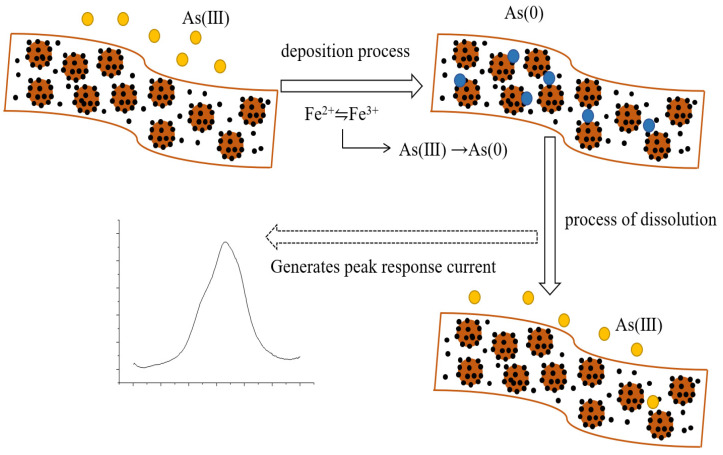
Experimental flow chart of an electrochemical sensor based on Fe_3_O_4_/MoS_2_ nanocomposites for the detection of arsenic ions.

**Figure 8 nanomaterials-13-02288-f008:**
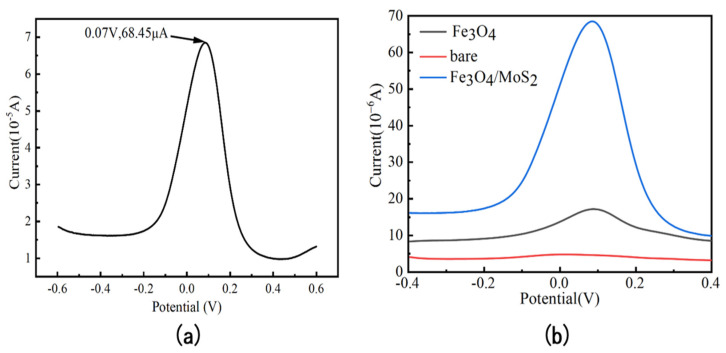
(**a**) Response of the Fe_3_O_4_/MoS_2_ nanocomposite modified electrode for the detection of 10 ppb arsenic ions after optimization of experimental conditions; (**b**) comparison of Fe_3_O_4_/MoS_2_, Fe_3_O_4_, modified electrode, and bare electrode for arsenic detection under the same experimental parameters.

**Figure 9 nanomaterials-13-02288-f009:**
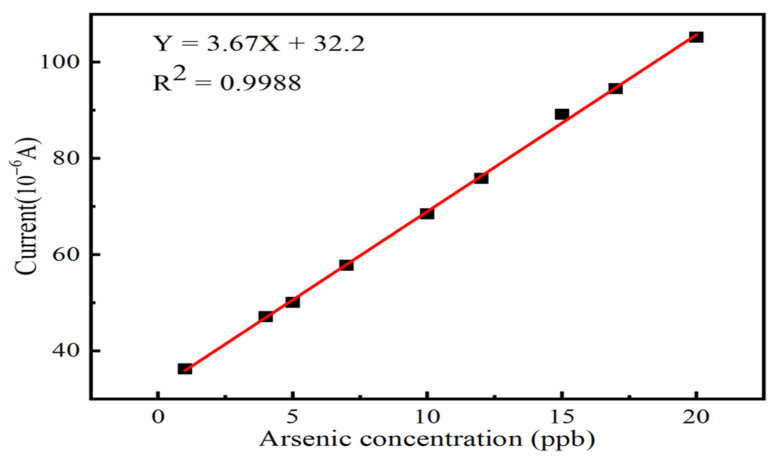
Peak response current versus arsenic ion concentration versus linear fit.

**Figure 10 nanomaterials-13-02288-f010:**
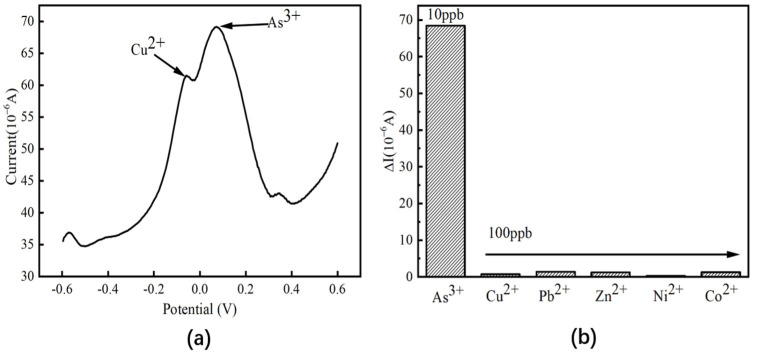
(**a**) SWV detection plot for arsenic detection in the presence of Cu^2+^ interference; (**b**) experimental graph of detection of interfering ions (Cu^2+^, Pb^2+^, Zn^2+^, Ni^2+^, Co^2+^).

**Figure 11 nanomaterials-13-02288-f011:**
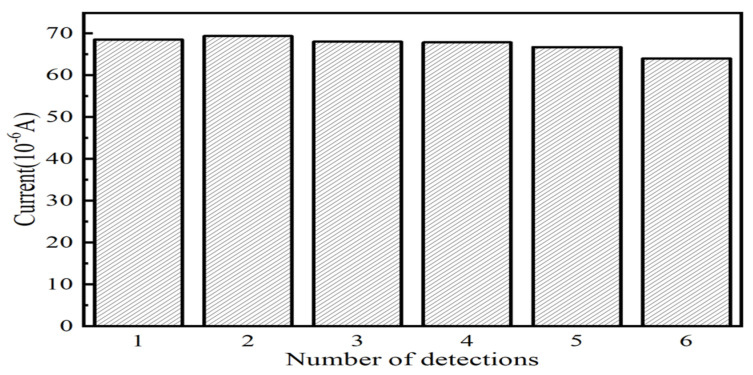
Comparison of peak currents of sensors with different numbers of repetitions.

**Figure 12 nanomaterials-13-02288-f012:**
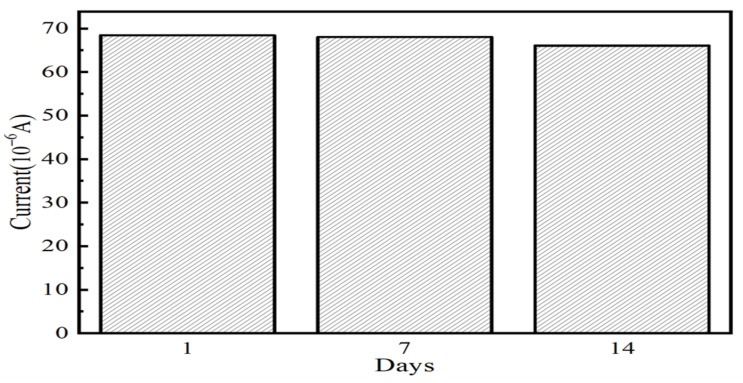
Stability testing of sensor arsenic detection by saving different time.

**Table 1 nanomaterials-13-02288-t001:** Comparative study of the performance of electrochemical sensors based on different materials for the detection of arsenic ions.

Electrode	Method	Sensitivity (µA/ppb)	Linear Range (ppb)	LOD (ppb)	Reference
NF(Au nano)	SWV	0.32	0.1–12.0	0.047	[31]
ERGO-AuNPs	ASV	0.16	0.75–374.6	0.20	[32]
AuNPs-Cfilms	ASV	0.026	1–100	0.55	[33]
np-Au	SWASV	0.60	0.5–15	0.0315	[34]
Pt_1_/MoS_2_	SWASV	3.31	0.5–8	0.05	[35]
AuNPs/α-MnO_2_	SWASV	0.828	1–10	0.019	[36]
rGO/MnO_2_	SWASV	0.175	0.1–50	0.05	[37]
MnO_2_/POT/rGO	DPV	0.00163	0.01–0.9	0.042	[38]
TiO_2_-GSE	LSV	1.10	10–80	10	[39]
Fe_3_O_4_/MoS_2_	SWV	3.67	1–20	0.70	This work

## Data Availability

Not applicable.

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
