# Peer review of "High Sensitivity Electrochemical As (III) Sensor Based on Fe3O4/MoS2 Nanocomposites"

_nanomaterials, 2023, doi:10.3390/nano13162288_

Round 1
Reviewer 1 Report
Dear Sir,
The article entitled "High sensitivity electrochemical As (III) sensor based on 2 Fe3O4/MoS2 nanocomposites” by the authors Haibing Hu et al. is quite interesting for the scientific community especially for researchers involved in the detection of heavy metal ion pollution in water. The paper describes a novel electrode based in Fe3O4/MoS2 composite that can detect low concentration of As(III) in water.
The article could be accepted after these modifications:
- In figure 7. Please put all the labels in English not in Chinese
- Improve English in the manuscript using more precise adjectives than “good”
Please replace adjective like good to more specific ones
Author Response
Thank you for your review
Point 1: In figure 7. Please put all the labels in English not in Chinese
Response 1: Thank you for your suggestion, the labelling in figure 7 has been changed to English.
Point 2: Improve English in the manuscript using more precise adjectives than “good”.
Response 2: Have replaced the use of good in the text with a better expression.

Reviewer 2 Report
The aim of this paper is preparation and study square wave voltammetry electrochemical detection of trace As(III) in water with a glassy carbon electrode modified with Fe3O4/MoS2 nanocomposites.
The results of this study will undoubtedly be of interest to specialists in the fields of nanomaterials and electrochemical sensors.
The topic of the paper is undoubtedly appropriate for "Nanomaterials". The paper contains new experimental results, and the quality of the presentation is adequate. The References List is sufficient, the Abstract and the Title reflect the contents adequately.
At the same time, the paper needs in some revisions:
1. Page 2, Line 92: The use of the term “medications” is inappropriate in this sentence.
2. Page 2, Line 97: The use of the term “Drugs” is inappropriate in this sentence.
3. In lines 93, 94, 100-102, 120, 129, the symbol “-” in chemical formulas must be replaced with the symbol “×”.
4. Page 3, Line 144: "trioxide" should be replaced with "oxide".
5. Page 4, Lines 172-176: The sentence is too long, it probably needs to be divided into two or more sentences.
6. Page 4, Lines 196-197: The conclusion “the calculated elemental molar ratio of Mo/S is approximately 1:2” does not follow from Figure 1d.
7. Page 4, Lines 198-204: The description of the basic principle of X-ray diffraction method does not belong to the “Results” section, moreover, it is well known, it should be deleted.
8. Page 4, Lines 214-217: The description of the basic principle of X-ray photoelectron spectroscopy does not belong to the “Results” section, moreover, it is well known, it should also be deleted.
9. Page 5, Line 219: "O2" should be replaced with "O" – this is the symbol of the element "oxygen".
10. Page 5, Figure 2a: The lower graph should be removed, since it is the sum of the two graphs above and does not carry any additional information.
11. Page 5, Figure 2b: Dimension after “Intencity” in the y-axis label is missing.
12. Page 7, Lines 264-268. The sentences are repeated.
13. Page 7, Lines 276-277: “nanoparticles. synergistic effect to improve the electrical conductivity.” is incomplete or unfinished sentence.
14. Page 7, Line 281: “106” should be changed to “106”.
15. Page 7, Lines 292-293: In the text fragment “the impedance of charge transfer was very low, enhancing electron transfer”, cause and effect are confused.
16. Neither the text nor the caption to Figure 4b indicate the magnitude of the direct electric current at which the frequency spectra of the electrochemical impedance were measured. Probably, these measurements were carried out at a current equal to 0.
17. Page 7, Figure 4b: The correct spelling of the dimension of the components of the impedance along the abscissa and ordinates is “Ohm”.
18. Page 7, Figure 4b: The real part of the impedance is Z¢.
19. Page 8, Equations (1)-(4): The unit “uA” should be changed to “mA”.
20. Page 8, Line 313: “Fe3O4@MoS2” should be replaced by “Fe3O4/MoS2” used in the text of the article.
21. Page 8 Lines 325-376: Probably, in the text of the article it is necessary to distinguish between the terms “buffer” and “supporting electrolyte”. The use of combinations “buffer solution (HCl)”, “sulphate buffer solution (H2SO4)” and “nitrate buffer solution (HNO3)” raises doubts – such solutions do not have buffer properties. A buffer solution (more precisely, pH buffer or hydrogen ion buffer) is an acid or a base aqueous solution consisting of a mixture of a weak acid and its conjugate base, or vice versa. Its pH changes very little when a small amount of strong acid or base is added to it. The addition of an excess of nonelectroactive ions (a supporting electrolyte) nearly eliminates the contribution of migration to the mass transfer of the electroactive species. In general, it simplifies the mathematical treatment of electrochemical systems.
22. Page 8, Line 328: “HAC” should be replaced with “HAc”.
23. Page 8, Line 330: “H2NO3” should be replaced with “HNO3”.
24. Figure 6, Figure 7, Figure 8a, Figure 11: The “Current” dimension must be replaced by the “mA” dimension after recalculating the current values.
25. Page 10, Figure 7: Chinese text needs to be translated into English.
26. Page 10, Line 388: “The” should be replaced with “the”.
27. Page 12, Line 449: “SO42–” should be replaced with “SO42–”.
28. Page 12, Line 449: “NO3–” should be replaced with “NO3–”.
29. Page 13, Figure 11a: Chinese text needs to be translated into English.
Comments on the Quality of English Language are listed in Comments and Suggestions for Authors.
Author Response
Thank you for your review. I have basically made modifications to your comments and have some personal thoughts. I hope to receive your feedback and correction again
Please refer to the attachment

Reviewer 3 Report
The paper presents research on the use of a sensor obtained by modifying a glassy carbon electrode with Fe3O4/MoS2 nanocomposites to detect traces of As(III) in water.
Overall the paper is well written and structured, but there are still a number of issues that should be resolved.
1. The purpose of the paper should be clearly stated at the end of the introductory part.
2. The novelty of the paper must be specified in the introductory part.
3. Section 2.1. Materials/chemicals should be corrected because the authors confuse drugs with reagents.
4.Section 2.2. Instrumentation must be completed with the types of devices that were used.
5. Figure 7 should be corrected, the Chinese writing should be removed.
6.Figure 11(a) must be redone because the legend is not understood.
7. The Conclusions section must be completed. The conclusions must be clear and contain a brief presentation of the obtained results.
